# Head and Neck Cancer Patients’ Survival According to HPV Status, miRNA Profiling, and Tumour Features—A Cohort Study

**DOI:** 10.3390/ijms24043344

**Published:** 2023-02-07

**Authors:** Ivana Šimić, Ksenija Božinović, Nina Milutin Gašperov, Mario Kordić, Ena Pešut, Luka Manojlović, Magdalena Grce, Emil Dediol, Ivan Sabol

**Affiliations:** 1Laboratory for Molecular Virology and Bacteriology, Ruđer Bošković Institute, 10000 Zagreb, Croatia; 2Laboratory for Cell Biology and Signalling, Ruđer Bošković Institute, 10000 Zagreb, Croatia; 3Department of Maxillofacial Surgery, Clinical Hospital Dubrava, 10000 Zagreb, Croatia; 4Department of Maxillofacial Surgery, Clinical Hospital Mostar, 88000 Mostar, Bosnia and Herzegovina; 5Clinical Department of Pathology and Cytology, Clinical Hospital Dubrava, 10000 Zagreb, Croatia

**Keywords:** HPV, microRNA, miR-21, HNC, tumour, survival

## Abstract

Head and neck cancers (HNC) are a heterogeneous group of tumours mainly associated with tobacco and alcohol use and human papillomavirus (HPV). Over 90% of all HNC are squamous cell carcinomas (HNSCC). Sample material from patients diagnosed with primary HNSCC (*n* = 76) treated with surgery as primary treatment at a single centre were assessed for HPV genotype, miR-9-5p, miR-21-3p, miR-29a-3p and miR-100-5p expression levels. Clinical and pathological data were collected from medical records. Patients were enrolled between 2015 and 2019 and followed-up until November 2022. Overall survival, disease-specific survival and disease-free survival were assessed and correlated with clinical, pathological, and molecular data. Kaplan-Meier and Cox proportional hazard regression was used to assess different risk factors. In the study, male gender, HPV-negative HNSCC (76.3%) mostly located in the oral region (78.9%) predominated. Most patients had stage IV cancer (47.4%), and the overall survival rate was 50%. HPV was found not to affect survival, indicating that in this population, classic risk factors predominate. The presence of both perineural and angioinvasion was strongly associated with survival in all analyses. Of all miRNAs assessed, only upregulation of miR-21 was consistently shown to be an independent predictor of poor prognosis and may thus serve as a prognostic biomarker in HNSCC.

## 1. Introduction

As a combined entity, head and neck cancer (HNC) is the eighth most common cancer globally with an incidence of more than 870,000 cases (lip, oral cavity, C00–C06; oropharynx, C09–C10; nasopharynx, C11; hypopharynx, C12–C13; larynx, C32) per year in the most recent GLOBOCAN 2020 data [1]. Approximately 90% of HNCs [2] are head and neck squamous cell carcinomas (HNSCCs), which derive from the mucosal epithelium in the upper aerodigestive tract [3]. Most HNSCCs, approximately 75%, share common risk factors including alcohol and tobacco use, while human papillomavirus (HPV) is the major risk factor for approximately 25% of HNSCCs [4]. Depending on the geographical region and culture, the mentioned risk factors have different importance for the incidence of HNSCC. For example, the decline in smoking prevalence in most high-income countries has facilitated the high prevalence of HNSCC in the US and Western Europe due to increased rates of oropharyngeal infection with carcinogenic HPV [3,5]. Thus, HPV-positive HNSCC is distinct from HPV-negative HNSCC regarding genetic, epigenetic, and protein expression profiles, epidemiological factors and clinical features, but the treatment for both groups is almost the same [6]. HPV is particularly associated with oropharyngeal squamous cell carcinoma (OPSCC), which affects the tonsils, base of the tongue, soft palate and uvula [5]. Also, HPV presence is an often-reported important factor when considering survival. Namely, five-year overall survival (OS) for HNC patients is poor (25–40%) [6]. However, some studies show that patients with HPV-positive OPSCC have improved OS, regardless of stage, to approximately 80% at five years [6,7]. While accurate for typical Western populations, these observations are not necessarily true for all populations [5,8].

The most frequently used staging system for HNC is the 8th edition of the American Joint Committee on Cancer (AJCC) Cancer Staging Manual [9]. The Manual classifies cancers by the size and extent of the primary tumour (T), involvement of regional lymph nodes (N), and the presence or absence of distant metastases (M), which are key factors that define prognosis and suitable treatment. According to the Manual, immunohistochemistry for p16 overexpression is a surrogate biomarker for HPV-mediated carcinogenesis and an independent positive prognosticator, not a direct detection of HPV. Besides the clinical and pathological TNM classification, other cancer-associated features like grade, invasion, lymph node yield (LNY), lymph node ratio (LNR), extranodal extension (ENE), depth of invasion (DOI), as well as patient-related factors, including the gender, age, and health status, the duration of symptoms, provide important prognostic information about patients’ outcome and response to treatment [9]. The lymph node ratio is defined as the ratio of positive lymph nodes to the total number of lymph nodes excised (lymph node yield, LNY) [10]. It is in some cases considered a better prognostic factor compared to N status because it reflects both the N status and the extent of the disease [10]. In addition to the above, there is some evidence that the anatomic location of lymph node involvement (referred to as lymph node levels and separated into five regions/groups, I-V) can affect the prognosis as well [11].

HNSCCs are a very heterogenetic entity where genetic and epigenetic changes play key roles. However, large-scale sequencing studies failed to identify particular oncogenes as drivers for HNSCC, while there were frequent disruptions of p53 and generally higher genomic instability [3]. HNSCC seems more likely to be an epigenetic disease rather than a genetic one, suggesting that molecules involved in epigenetic changes, such as microRNAs, may be promising biomarkers in these cancers [6]. MicroRNAs (miRNAs, miRs) are endogenous, small non-coding RNAs (21–25 nucleotides) which play an important role in several biological functions, including cell proliferation, apoptosis, tumour growth, and metastasis [12,13]. MiRNAs have been shown to function as oncogenes or tumour suppressor genes depending on the genes and pathways they target [14,15,16]. Studies have shown that miRNA profiles in cancer can be used to discriminate between different developmental lineage and differentiation statuses and predict patient survival and treatment response [17].

The aim of this study was to correlate HPV status, miRNAs, tumour features, and features of HNSCC patients to identify their association with outcomes, especially in a population with a stronger influence of classical HNSCC risk factors, i.e., tobacco and alcohol use.

## 2. Results

### 2.1. Patient and Tumour Characteristics

Over the patient enrolment period (2015–2019), sample material was collected from 87 patients. However, in six cases, the material was not from the primary tumour; for three patients, the histopathologic diagnosis indicated that the tumour was not squamous cell carcinoma, and two patients had inoperable tumours. Thus 11 patients were excluded from further analyses.

Men prevailed in the study (73.7%, 56/76) in comparison with women (26.3%, 20/76). The mean age of all patients was 61.7 ± 11.6 (range 31–87, median 61). Female patients were, on average, slightly older (63.4 ± 14.1) than male patients (61.1 ± 10.6). Approximately one-third of patients were never smokers and never drinkers (34.2%, 26/76), while more than 60% of patients reported tobacco and alcohol use (Table 1).

Most tumours were in the oral region (78.9%, 60/76), while others were in the oropharyngeal region (21.1%, 16/76). Among the anatomical sub-localizations, the gingiva, oral tongue, and sublingual area predominated (Table 1). The tongue and sublingual area were most commonly affected in men, while the gingiva was most commonly affected in women.

According to clinical TNM (cTNM), most patients presented with stage IV cancer (47.4%). No patient had distant metastasis at the time of surgical treatment (M = 0); however, 17 patients developed metastasis during the follow-up (Table 1). Most patients with subsequent distant metastases presented with stage IV disease (11/17) and were HPV-negative (12/17). Clinically most patients were classified as T2, followed by T4a, T3 and T1. However, local lymph node involvement was seen in more than half of the patients (cN > 0, 53.9%, 41/76).

Samples were classified according to HPV status into two groups: 18 (23.7%) samples with HPV-positive and 58 samples with HPV-negative HNSCC (76.3%). The HPV-positive patients were only slightly younger than the HPV-negative group on average (59.6 ± 15.5 vs. 62.4 ± 10.1 years), and the difference was not statistically significant (*t*-test, *p* = 0.4749). Among HPV-positive tumours HPV16 was found in most cases (77.7%, 14/18). Almost 60% of patients underwent surgery as the only treatment, while the rest of the patients also subsequently received radiotherapy (Table 1). Only four patients received chemoradiotherapy, and they were grouped with those receiving radiotherapy and not reported separately.

**Table 1 ijms-24-03344-t001:** Head and neck patient characteristics.

Characteristics		Total (*n* = 76)	HPV-Positive (*n* = 18)	HPV-Negative (*n* = 58)
Gender	F	20 (26.3%)	4 (22.2%)	16 (27.6%)
	M	56 (73.7%)	14 (77.8%)	42 (72.4%)
Age	Mean ± SD	61.7 ± 11.6	59.6 ± 15.5	62.4 ± 10.1
	Median	61	61.5	61
	Range	31–87	31–87	42–85
	0–64	46 (60.5%)	11 (61.1%)	35 (60.3%)
	65+	30 (39.5%)	7 (38.9%)	23 (39.7%)
Lifestyle	NSmND	26 (34.2%)	8 (44.4%)	18 (31%)
	Sm	5 (6.6%)	1 (5.6%)	4 (6.9%)
	SmD	45 (59.2%)	9 (50%)	36 (62.1%)
Tumour site	Oral cavity	60 (78.9%)	12 (66.7%)	48 (82.8%)
	*gingiva*	17 (22.4%)	3 (16.7%)	14 (24.1%)
	*oral tongue*	16 (21.1%)	2 (11.1%)	14 (24.1%)
	*sublingual area*	16 (21.1%)	5 (27.8%)	11 (19%)
	*retromolar*	9 (11.8%)	2 (11.1%)	7 (12.1%)
	*buccal mucosa*	2 (2.6%)	0 (0%)	2 (3.4%)
	Oropharynx	16 (21.1%)	6 (33.3%)	10 (17.2%)
	*base of tongue*	8 (10.5%)	1 (5.6%)	7 (12.1%)
	*tonsil*	7 (9.2%)	5 (27.8%)	2 (3.4%)
	*posterior pharyngeal wall*	1 (1.3%)	0 (0%)	1 (1.7%)
cT stage	1	6 (7.9%)	2 (11.1%)	4 (6.9%)
	2	26 (34.2%)	5 (27.8%)	21 (36.2%)
	3	21 (27.6%)	5 (27.8%)	16 (27.6%)
	4a	23 (30.3%)	6 (33.3%)	17 (29.3%)
cN stage	0	35 (46.1%)	7 (38.9%)	28 (48.3%)
	1	20 (26.3%)	5 (27.8%)	15 (25.9%)
	2	17 (22.4%)	5 (27.8%)	12 (20.7%)
	3	4 (5.3%)	1 (5.6%)	3 (5.2%)
Overall clinical stage	Stage I–II	21 (27.6%)	2 (11.1%)	19 (32.8%)
	*I*	3 (3.9%)	0 (0%)	3 (5.2%)
	*II*	18 (23.7%)	2 (11.1%)	16 (27.6%)
	Stage III	19 (25%)	6 (33.3%)	13 (22.4%)
	Stage IV	36 (47.4%)	10 (55.6%)	26 (44.8%)
	*IVa*	32 (42.1%)	9 (50%)	23 (39.7%)
	*IVb*	4 (5.3%)	1 (5.6%)	3 (5.2%)
HPV	Positive	18 (23.7%)	18	
	Negative	58 (76.3%)		58
Therapy	Surgery	45 (59.2%)	10 (55.6%)	35 (60.3%)
	Surgery + RT	31 (40.8%)	8 (44.4%)	23 (39.7%)
Follow-up	Survival overall	38 (50%)	10 (55.6%)	28 (48.3%)
	No evidence of disease	36 (47.4%)	9 (50%)	27 (46.6%)
	Alive with disease	2 (2.6%)	1 (5.6%)	1 (1.7%)
	Dead of other causes	23 (30.3%)	3 (16.7%)	20 (34.5%)
	Dead of disease	15 (19.7%)	5 (27.8%)	10 (17.2%)
Follow-up events	Recurrence	13 (17.1%)	4 (22.2%)	9 (15.5%)
	Distant metastasis	17 (22.4%)	4 (22.2%)	13 (22.4%)
	Secondary malignancy	16 (21.1%)	3 (16.7%)	13 (22.4%)
	Median follow-up time (months)	45.5	40.5	46

F—female, M—male, SD—standard deviation, NSmND—never smoker/never drinker, Sm—smoker, SmD—smoker and drinker, c—clinical, HPV—human papillomavirus, RT—radiotherapy.

Results of the histopathology assessment are presented in Table 2. Histopathological T stage followed a clinical pattern with T2 being the most represented, followed by T4a, T3 and T1. The concordance of cT and pT was high (Kappa interrater agreement 0.68; 95% CI 0.55–0.81), but it was lower for cN and pN classification (Kappa = 0.48; 95% CI 0.56–0.61). However, if patients with pNX were considered to be N0 since the tumour was considered small and surgical intervention minimal, then the concordance was again high (Kappa = 0.65; 95% CI 0.52–0.77). Pattern congruence also exists between clinical and pathological overall stage (Kappa = 0.74; 95% CI 0.61–0.87), with most staged identically while nine cases were up-staged on pathological classification and three down-staged. Histologically, most patients had grade II tumours (46.1%, 35/76). Angioinvasion or perineural invasion was present in the majority of tumours (65.8%, 50/76). There were no signs of any invasion in 26 patients. In the majority of patients (53.9%), more than 21 lymph nodes were excised. Half of the patients’ lymph nodes were not involved (Table 2).

### 2.2. MiRNA Expression

Fold changes of assessed miRNA molecules (miR-9-5p, miR-21-3p, miR-29a-3p and miR-100-5p) compared to the normal sample pool across patient subgroups are shown in Table 3. Consistent upregulation of miR-21 can be seen across the study population and all subgroups (Figure 1). MiR-9 was slightly more increased in HPV-positive subsets, while miR-29a-3p and miR-100-5p were downregulated across the groups. These findings are consistent with the HNSCC cohort of The Cancer Genome Atlas (*TCGA*) [18] (Appendix A). Expression of other assessed miRNA molecules that could not be assessed in the complete sample population due to lack of material is presented in Appendix A.

### 2.3. Overall Survival

The median follow-up was 45.5 months, with a range from 0 to 82 months for the whole population. The median follow-up time of living patients was 61 months. Half of the patients were alive (38/76, 50%), and most of those living (36/38, 94.7%) had no evidence of disease, while 50% (38/76) of patients died (Table 1). The overall survival rate was higher among women (70% vs. 42.9%). There is no considerable difference in survival rate between those with HPV-positive (10/18, 55.5%) and those with HPV-negative tumours (28/58, 48.3%). The overall survival rate in relation to the stage was as follows: 66.7% (14/21) stage I + II, 36.8% (7/19) stage III and 47.2% (17/36) stage IV.

The estimated five-year survival (Kaplan-Meier analysis) was 75.6% for stage I + II, 33.8% for stage III and 44.9% for stage IV. However, the difference between survival curves when staging was done according to clinical and pathological TNM variables was not statistically significant (*p* = 0.141 and *p* = 0.127, respectively) (Figure 2).

Kaplan-Meier analysis was performed for all analysed parameters according to OS, DSS, and DFS survival and all curves are shown in Appendix A.

Interestingly, only the presence of invasion (Figure 3) and the high expression of miR-106b were consistently associated with significantly different survival curves for all three outcomes assessed (overall survival - OS, disease specific survival - DSS and disease free survival–DFS). In all cases finding both perineural and angioinvasion was associated with worse outcomes (OS *p* = 0.014; DSS *p* = 0.007; DFS *p* = 0.005; Figure 3). Not all patients could be assessed for miR-106b; however, for those available (*n* = 69), the survival curves were remarkably different (OS *p* = 0.020; DSS *p* = 0.024; DFS *p* = 0.026; Appendix A). Notable is the lymph node yield analysis where the worst performing patients had 1–10 lymph nodes removed for analysis, while the best outcome was in those with 0 nodes dissected (OS *p* = 0.006; DSS *p* = 0.031; DFS *p* = 0.026). Clean resection margins were significant for DSS (*p* = 0.036) but not for OS (*p* = 0.138) (Appendix A).

### 2.4. Risk Factors in HNSCC Patients

To assess the influence of clinical parameters as well as measured expression levels of miRNA molecules, univariate and multivariate Cox proportional hazard regression models for the OS, DSS and DFS were made. All univariate results are presented in Appendix A.

Multivariate models were created by entering variables with *p* ≤ 0.1 in univariate analysis, and the models are presented in Table 4. All three models were significant overall (OS, *p* < 0.001; DSS, *p* = 0.028; DFS, *p* < 0.001); however, for the DSS model, no variables were individually significant. For OS, gender, presence of perineural and angioinvasion, tumour resection margin, lymph node yield and the expression of miR-21 were independent prognostic factors. MiR-21 was also found to be significant in the DFS model.

Having noted the significant association of miR-21 in multivariate models as well as potential trends of miR-106b seen in the univariate analysis, we further investigated the effect of these miRNAs on survival across different stage categories using Kaplan-Meier analysis (Figure 4). High expression of either miRNA strongly affected the survival of patients with stage III or stage IV cancer, while the survival curves of stage I–II patients were almost not affected, especially for miR-106b (panels B, D-F green lines). Analysis of other miRNAs (-9, -29a and -100) did not yield significant differences and was not shown.

Association of individual miRNA expression (miR-9, miR-21, miR-29a and miR-100) with overall survival were evaluated with ROC curve analysis. However, the resulting AUC values were poor (between 0.52 and 0.57; Appendix A).

## 3. Discussion

Over the last few decades, there has been substantial improvement in HNSCC understanding, diagnosis, staging and treatment [19]. The management of the patients is based on histologic parameters such as TNM staging and tumour grading. However, there is a need for new biomarkers in order to improve patients’ prognosis and treatment [20].

HNSCC is one of the most aggressive cancers due to advanced disease at the time of diagnosis [13]. Most of the patients in the current study indeed presented with stage IV cancer (47.4%). Nevertheless, this number of stage IV patients’ five-year mortality was 50%. The mortality rate of the patients was below the five-year European average age-standardized mortality reported in 2015 (65.6%) [21]. Although tobacco and alcohol were the predominant risk factors here, HPV is often also associated with HNSCC, especially in the case of OPSCC [22]. However, the majority of the tumours in this study were located in the oral region (78.9%), which is not the predilection site for HPV infection, and this might explain the diminished impact of HPV on survival in the present study. The HPV effect on OS in non-OPSCC is controversial [6], ranging from no prognostic benefit [23] to a good prognostic factor [24,25]. HPV16 was the most common type, which is in accordance with previous studies [5,26]; however, HPV-positive patients were only slightly younger in comparison with HPV-negative patients and the presence of HPV as determined by mRNA and DNA analysis was also not significantly associated with survival unlike in other more Western populations [27,28]. A similar non-existent HPV-associated survival benefit was observed in a previous retrospective analysis on an unrelated set of samples (*n* = 99 FFPE) [29], suggesting that different populations might have different contributions of HPV as a protective factor.

Approximately 40% of HNSCC were diagnosed in elderly patients (65+ years) (Table 1), and this observation is in line with the current literature [21]. However, elderly patients frequently have comorbidities, which make them susceptible to other diseases and death [30], and this is also evident here, where more patients died from other causes (30.3%) than from HNSCC (19.7%) (Table 1).

Our study confirmed earlier findings that the male population is more prone to both HPV-positive and HPV-negative HNSCC [31]. Possible reasons for such distribution of HPV-positive HNSCC could be associated with hormonal differences as well as different HPV transmission rates between genders, although the latter is still ambiguous according to the literature [31]. The observed gender inequality could also be explained by the fact that men are more likely than women to report increased numbers of sexual partners [5]. The culturally and traditionally conditioned presence of high-risk factors, such as tobacco and alcohol consumption, is more associated with the male population and could explain the predominance of males among those with HPV-negative HNSCC [3]. However, the uneven representation of women and men in this study could lead to potential bias. Thus, the weak significant association of gender with overall survival on multivariate Cox regression should be interpreted with caution (male vs female, HR 3.1 95% CI 1.1–8.7, *p* = 0.027, Table 4). There was no difference in stage at patient presentation between genders (chi square test, *p* = 0.962; Appendix A), thus, health awareness cannot be the cause for survival differences between gender. On the other hand, men were more subject to classic risk factors (smoking and drinking), and the difference was striking (15.0% female vs 75% men reporting alcohol and tobacco use, chi-square *p* < 0.001; Appendix A), which could easily explain the gender survival rate differences.

One significant prognostic factor was the presence or absence of perineural and angioinvasion. According to our observation, the worst survival could be observed in patients who had a combined type of invasion, while patients without any invasion had the best survival. The presence of invasion was strongly associated with survival in all versions of Kaplan-Meier and Cox analysis for all three examined outcomes (OS, DSS and DFS) suggesting a strong robustness of the association (Figure 3). Cox models also implied strong influence of lymph node yield where patients with 1–10 evaluated nodes had a relatively high risk of death in overall survival (HR 11.6 95% CI 1.2–9.7) and disease-free survival analysis (HR 10.0 95% CI 1.8–54.2) (Table 4) compared with patients where no lymph nodes were dissected. This observation can be biased due to the extent of the disease, where small tumours are minimally excised and where lymph nodes are intentionally not evaluated. Interestingly, clinical and pathological N stage or the actual number of positive lymph nodes found in the patient material or high LN ratio were never associated with survival on univariate Cox regression analysis (Appendix A) or Kaplan-Meier analysis (Appendix A). Precise, numerical rather than a descriptive determination of the resection margins by the pathologist was implicated in survival (OS and DFS). Patients who received only a descriptive designation of resection margin (“clean”) had a higher hazard ratio (OS HR 4.5, 95% CI 1.5–13.6) compared with patients where resection margin was 5 mm or more, which was even worse than the hazard ratio of patients with known close margins (OS HR 1.9, 95% CI 0.7–4.7) (Table 4).

In accordance with some previous studies [13,15], our study showed that elevated miR-21 levels were an independent predictor of poor prognosis (Table 4), with especially strong differences in stage IV cancer (Figure 4A–C). Depending on the tumour type, anatomic site or cellular context, miR-9 can act as an oncomiR or oncosuppressor [13]. In this study, we observed an increasing trend of miR-9-5p in HPV-positive tumours (Figure 1). However, possibly due to the limited number of samples from different sub-locations, any effects on survival were not significant. Additionally, a high expression level of miR-100 was correlated with poorer overall survival (Appendix A), similar to some other studies [18,32]. However, miR-100 did not reach statistical significance in Kaplan-Meier or Cox proportional regression analysis. Although it was not possible to detect miR-106b in the whole set of samples, decreased miR-106b was correlated to better survival, and this finding was statistically significant in some cases of univariate Cox analysis (Appendix A) as well as in Kaplan-Meier analysis when combined with tumour stage (Figure 4D–F).

The limitations of our study include the relatively small number of patients and potential but unavoidable gender bias. Other limitations include the possible reporting biases and underreporting of important cancer features as evidenced in the variability of pathologists’ reports, with some describing resection margins as narrow or wide rather than providing precise measurements in some cases, some explicitly noting different invasions or the presence of ENE. Due to the inconsistencies in reporting the depth of invasion, it was completely impossible to assess this as a separate parameter in most cases. Unfortunately, almost no p16 staining was performed, except in 4 out of 76 cases, limiting the assessment of this useful biomarker in our study population. The implementation and completion of a uniform report for HNC would reduce this issue.

## 4. Material and Methods

### 4.1. Patients and Tumour Samples

All patients in this cohort study underwent surgery as primary treatment for HNSCC performed at the Department of Maxillofacial Surgery, Clinical Hospital Dubrava, University of Zagreb, Croatia, between 2015 and 2019 and followed-up until November 2022. Patients with a diagnosis of oral cancer (O; gingiva, retromolar area, oral tongue, sublingual area—excluding base of the tongue, buccal mucosa), as well as oropharyngeal cancer (OP; base of the tongue, tonsil, posterior pharyngeal wall) were included. In our previous study [33], a subset of patients was assessed by high-throughput methods for miRNA profiling. However, at that time, the patient follow-up was too short to make observations regarding outcomes. Within the current study, we collected follow-up information and included additional patients treated in the intervening period.

A total of 76 patients, 20 women (age range 32–87 years) and 56 men (age range 31–85 years), were included in the current study. Patients’ data from previously collected (*n* = 59) and newly enrolled patients (*n* = 17) are shown in Appendix A. For the current study additional detailed information including clinical tumour, node, and metastasis (cTNM) status, pathological TNM (pTNM) status, histological grade, presence of histopathologically assessed angioinvasion or perineural invasion or their combination, tumour involvement of surgical margins (and the distance to the margin), lymph node yield (LNY), positivity and ratio (LNR), presence of extranodal extension (ENE), postoperative treatments and survival information including disease recurrence (Table 1 and Table 2) were collected from hospital databases and medical records for all included patients. Tumours were staged by using the 8th edition of the AJCC Cancer Staging Manual [9]. Since p16 information was not available, cases were staged according to the p16 negative guidelines. Separate staging is shown for clinical and pathological TNM classification (Table 1 and Table 2). Furthermore, alcohol and tobacco consumption as risk factors were reviewed; however, pack/year or detailed alcohol consumption data were not available.

Informed consent was obtained from all patients, and the study (Epic-HNSCC project No 4758) was approved by the Clinical Hospital Dubrava Bioethics Committee (EP-KBD-10.06.2014) and the Ruđer Bošković Institute Bioethics Committee (BEP-3748/2-2014).

### 4.2. RNA and DNA Extraction and HPV Testing

Detailed procedures were previously described [33] and were also applied to the newly enrolled patients. Briefly, DNA and RNA were isolated from two separate tumour tissue samples using an EZ1 DNA Tissue kit (Qiagen, Hilden, Germany), GenElute-E Single Spin Tissue DNA Kit (Sigma-Aldrich, St. Louis, MO, USA) and miRNeasy Mini Kit (Qiagen, Hilden, Germany) for the isolation of DNA and RNA, respectively. The total RNA and DNA concentration of samples was measured using a NanoPhotometer (Implen, München, Germany).

Polymerase chain reaction (PCR) for HPV was performed with consensus HPV-specific primers PGMY, GP5+/GP6+, LC, HPV16 and HPV18 type-specific primers as described previously [33] and detailed in the Appendix A. Beta-globin PCR amplification was used as an internal control. The E6 mRNA analysis was performed on HPV16 DNA-positive samples as previously described [33]. Only HPV16 mRNA-positive and HPV18-positive samples were considered likely to be HPV associated.

### 4.3. MicroRNA Quantitation

The expression of miRNA targets was assessed using TaqMan advanced miRNA Assays (Applied Biosystems, Waltham, MA, USA) as described before [33]. cDNA was prepared from total RNA from samples using the TaqMan Advanced miRNA cDNA Synthesis Kit (Applied Biosystems, Waltham, MA, USA). Expression of miR-9-5p, miR-21-3p, miR-29a-3p and miR-100-5p were selected for further investigation in the current study. miR expression was normalised using the average of miR-16-5p, miR-181-5p and miR-191-5p as an internal reference control. Quantitative RT–PCR reaction was performed using CFX96 Touch Real-Time PCR Detection System (BioRad, Hercules, CA, USA) for newly collected samples. Cycle conditions and reaction volumes used for poly(A) tailing, adaptor ligation, reverse transcription, miR-Amp reaction and qPCR were according to the manufacturer’s instructions. A pool of cDNA from three healthy tonsil samples was used as the referent sample [33]. The 2^−ΔΔCt^ method was used to calculate the fold changes. Due to a lack of material in some cases miR-106b-5p, miR-143-3p, miR-145-5p, and miR-199b could not be analysed for all samples, but old data were re-evaluated in the context of additional collected clinical, as well as survival data. Furthermore, to avoid issues with proper controls for normalization of the 2^−ΔΔCt^ calculations, obtained expression data were also analysed as single 2^−ΔCt^ versus internal reference miRNAs only. This additional 2^−ΔCt^ relative expression analysis is only presented as Appendix A.

### 4.4. Statistical Analysis

All statistical analyses were performed using the MedCalc software (version 20.111). Patients were dichotomised by the median expression of each studied miRNA into above and below median groups. Groups were also divided into above or below 65 years of age. Variables with levels consisting of few cases were grouped, i.e., stage I (*n* = 3) and stage II (*n* = 18) cancers were combined, as well as different sublevels of stage IV (IVa, *n* = 32 and IVb, *n* = 4) to allow more meaningful comparisons. Overall survival (OS—patients who died of any cause), disease-specific survival (DSS—patients who died of the main disease) and disease-free survival (DFS—any recurrence, distant metastasis or death of any reason) were recorded. Follow-up time in months was calculated from the date of enrolment (surgery) to the registered time of death or last registered follow-up in case of OS or DSS. Follow-up time in months from enrolment to disease recurrence, metastasis or death of any cause was considered for disease-free survival assessment. Individual parameters were assessed for influence on survival using the Kaplan-Meier method and log-rank test. Cox proportional hazard regression was used to assess the impact of examined parameters on the outcomes using the same time and event criteria as for the Kaplan-Meier method. Both univariate and multivariate models, with relevant variables, were constructed. Final multivariate models were constructed using non-redundant variables with *p* < 0.1 in univariate analysis (i.e., if both clinical and pathological TNM stage variables had *p* < 0.1, only the clinical one was included in the multivariate model). The performance of different miRNAs was also analysed with ROC curve analysis for overall survival. All *p*-values below 0.05 were considered statistically significant.

## 5. Conclusions

HNSCC is a highly aggressive cancer with nearly 50% of patients having the advanced disease at the time of diagnosis and a 50% mortality rate. Tobacco and alcohol continue to play a leading role as risk factors in HNSCC within the Croatian population, especially among men, while HPV seems to have a lower impact. However, a wider sample panel and p16 testing would be useful to draw a definitive conclusion. On the other hand, our results indicate that certain miRNAs, such as miR-21 and miR-106b, might serve as useful prognostic biomarkers in HNSCC, in addition to tumour features like perineural infiltration, angioinvasion and lymph node yield.

## Figures and Tables

**Figure 1 ijms-24-03344-f001:**
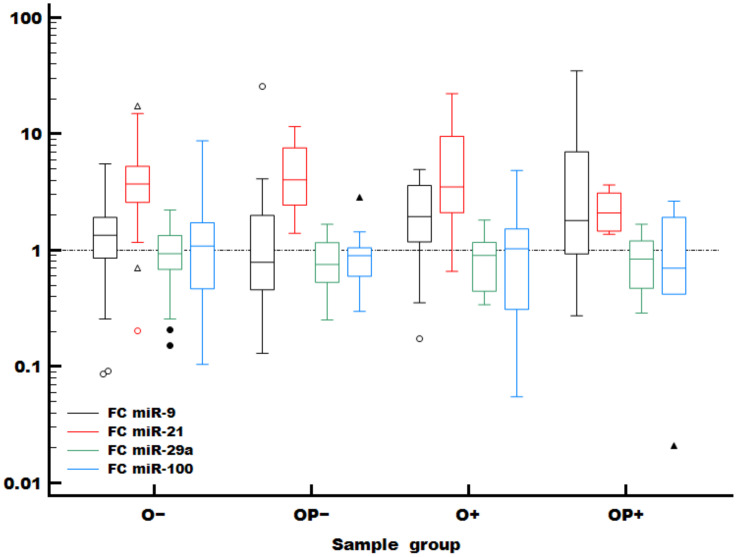
Relative expression (FC—fold change vs. referent normal control sample pool) of studied miRNA molecules in patient subgroups. The horizontal reference line is set to 1 (no change). O—oral, OP—oropharyngeal, O+, OP+—HPV-positive oral and oropharyngeal tumours, O−, OP−—HPV-negative oral and oropharyngeal tumours. Outlying values are indicated as individual data points (full and empty circles and triangles).

**Figure 2 ijms-24-03344-f002:**
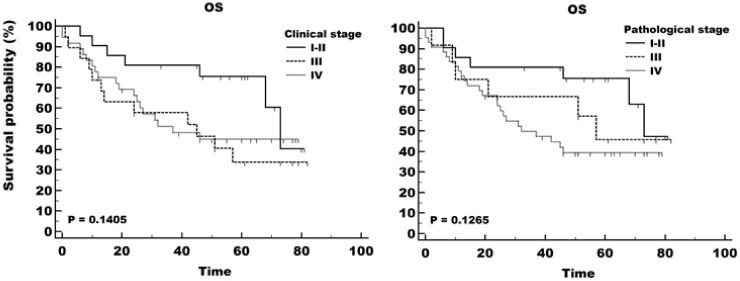
Overall survival (OS) of patients with different stages of head and neck cancer. Staging was done according to the clinical TNM parameters on the left and pathological TNM parameters on the right-side panel.

**Figure 3 ijms-24-03344-f003:**
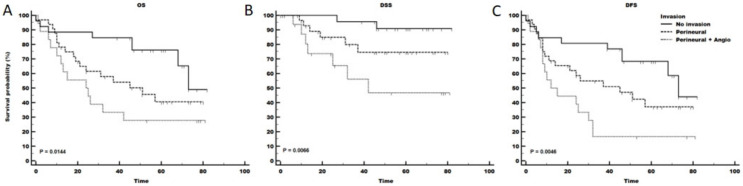
Survival ((**A**) panel—OS, overall survival; (**B**) panel—DSS, disease-specific survival; (**C**) panel- DFS, disease-free survival) of patients with different types of invasions. The same legend applies to all three panels. The survival curves were statistically significantly different regardless of the survival endpoint assessed. *p* value of the Kaplan-Meier analysis log-rank test is presented in the bottom left corner of each plot.

**Figure 4 ijms-24-03344-f004:**
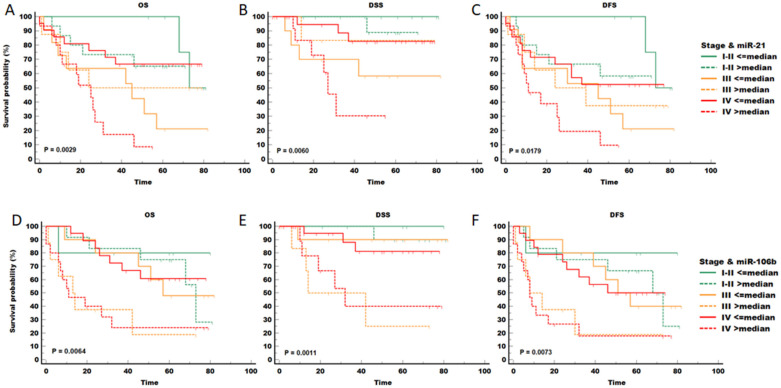
Association of miR-21 and miR-106b expression with survival of patients with different stages of head and neck cancer. Cancer severity colour coded: stage I–II green, stage III orange, and stage IV red. Dotted lines indicate patients with above median expression of indicated miRNA (miR-21 panels (**A**–**C**), miR-106b panels (**D**–**F**)). In both cases, red full lines and red dotted lines separate strongly with stage IV cancer being also the most prevalent in the study population. OS—overall survival, DSS—disease-specific survival, DFS—disease-free survival.

**Table 2 ijms-24-03344-t002:** Head and neck patient histopathological data.

Characteristics		Total (*n* = 76)	HPV-Positive (*n* = 18)	HPV-Negative (*n* = 58)
pT stage	1	6 (7.9%)	2 (11.1%)	4 (6.9%)
	2	30 (39.5%)	5 (27.8%)	25 (43.1%)
	3	15 (19.7%)	3 (16.7%)	12 (20.7%)
	4a	25 (32.9%)	8 (44.4%)	17 (29.3%)
pN stage	X	11 (14.5%)	5 (27.8%)	6 (10.3%)
	0	27 (35.5%)	3 (16.7%)	24 (41.4%)
	1	8 (10.5%)	1 (5.6%)	7 (12.1%)
	2	21 (27.6%)	5 (27.8%)	16 (27.6%)
	3	9 (11.8%)	4 (22.2%)	5 (8.6%)
Overall pathological stage	Stage I–II	21 (27.6%)	3 (16.7%)	18 (31%)
	*I*	4 (5.3%)	1 (5.6%)	3 (5.2%)
	*II*	17 (22.4%)	2 (11.1%)	15 (25.9%)
	Stage III	12 (15.8%)	3 (16.7%)	9 (15.5%)
	Stage IV	43 (56.6%)	12 (66.7%)	31 (53.4%)
	*IVa*	34 (44.7%)	8 (44.4%)	26 (44.8%)
	*IVb*	9 (11.8%)	4 (22.2%)	5 (8.6%)
Grade	1	29 (38.2%)	5 (27.8%)	24 (41.4%)
	2	35 (46.1%)	8 (44.4%)	27 (46.6%)
	3	12 (15.8%)	5 (27.8%)	7 (12.1%)
Resection margins	Clean (≥5 mm)	25 (32.9%)	6 (33.3%)	19 (32.8%)
	Clean (NS)	21 (27.6%)	4 (22.2%)	17 (29.3%)
	Close (<5 mm)	30 (39.5%)	8 (44.4%)	22 (37.9%)
Invasion	No invasion	26 (34.2%)	6 (33.3%)	20 (34.5%)
	Perineural	32 (42.1%)	8 (44.4%)	24 (41.4%)
	Perineural + Angio	18 (23.7%)	4 (22.2%)	14 (24.1%)
Lymph node yield	0	10 (13.2%)	4 (22.2%)	6 (10.3%)
	1–10	8 (10.5%)	1 (5.6%)	7 (12.1%)
	11–20	17 (22.4%)	2 (11.1%)	15 (25.9%)
	21+	41 (53.9%)	11 (61.1%)	30 (51.7%)
Lymph node positivity	0	38 (50%)	8 (44.4%)	30 (51.7%)
	1–2	22 (28.9%)	5 (27.8%)	17 (29.3%)
	3–4	9 (11.8%)	3 (16.7%)	6 (10.3%)
	5+	7 (9.2%)	2 (11.1%)	5 (8.6%)
Lymph node ratio	≤0.05	35 (46.1%)	7 (38.9%)	28 (48.3%)
	>0.05	31 (40.8%)	7 (38.9%)	24 (41.4%)
	No lymph nodes evaluated	10 (13.2%)	4 (22.2%)	6 (10.3%)
Extranodal spread	No	24 (31.6%)	4 (22.2%)	20 (34.5%)
	Yes	14 (18.4%)	6 (33.3%)	8 (13.8%)
	No positive LNs	38 (50%)	8 (44.4%)	30 (51.7%)

p—pathological, X—regional lymph nodes cannot be assessed, NS—not specified, NA—not applicable, LN—lymph node.

**Table 3 ijms-24-03344-t003:** Relative expression (median and IQR, fold change versus normal control sample) of miRNA molecules in the study population and subgroups.

miRNA	Total (*n* = 76)	HPV-Negative (*n* = 58)	HPV-Positive (*n* = 18)	O Total (*n* = 60)	OP Total (*n* = 16)	O− (*n* = 48)	O+ (*n* = 12)	OP− (*n* = 10)	OP+ (*n* = 6)
miR-9	1.4 (0.8–2.6)	1.2 (0.7–1.9)	1.8 (1–4.2)	1.4 (0.9–2.6)	0.9 (0.5–3.1)	1.3 (0.9–1.9)	2 (1.2–3.6)	0.8 (0.5–2)	1.8 (0.9–7)
miR-21	3.6 (2.3–5.5)	3.8 (2.5–5.5)	3 (1.7–7.5)	3.7 (2.5–5.6)	3.3 (1.9–4.3)	3.7 (2.6–5.3)	3.5 (2.1–9.7)	4 (2.4–7.6)	2.2 (1.5–3.1)
mir-29a	0.9 (0.6–1.3)	0.9 (0.7–1.3)	0.9 (0.5–1.2)	0.9 (0.7–1.3)	0.8 (0.5–1.2)	0.9 (0.7–1.3)	0.9 (0.4–1.2)	0.8 (0.5–1.2)	0.9 (0.5–1.2)
miR-100	1 (0.4–1.6)	1 (0.5–1.6)	0.9 (0.3–1.6)	1 (0.4–1.7)	0.9 (0.5–1.2)	1.1 (0.5–1.7)	1 (0.3–1.5)	0.9 (0.6–1.1)	0.7 (0.4–1.9)

O, oral region; OP, oropharyngeal region; O−, HPV-negative tumour in oral region; O+, HPV-positive tumour in oral region, OP−, HPV-negative tumour in oropharyngeal region; OP+, HPV-positive tumour in oropharyngeal region.

**Table 4 ijms-24-03344-t004:** Multivariate Cox hazard regression models.

	Overall Survival		Disease-Specific Survival		Disease-Free Survival	
Events	38/76		14/69 *		43/76	
Overall model fit	*p* < 0.001		*p* = 0.028		*p* < 0.001	
	HR (95% CI)	*p*	HR (95% CI)	*p*	HR (95% CI)	*p*
Age 65+ vs <65	-	-	-	-	3 (1.4–6.4)	0.0042
Gender M vs F	3.1 (1.1–8.7)	0.027	-	-	-	-
Sm vs NSmND	2.1 (0.4–10.7)	0.360	-	-	-	0
SmD vs NSND	1.7 (0.7–4.1)	0.206	-	-	-	0
Stage III vs I–II	1.6 (0.5–4.9)	0.419	3.9 (0.4–38.6)	0.244	1.2 (0.5–3.3)	0.674
Stage IV vs I–II	1.1 (0.4–3.5)	0.839	3.5 (0.4–31.9)	0.265	1.4 (0.6–3.7)	0.464
Perivascular vs no invasion	2.8 (0.8–9.6)	0.099	3.9 (0.5–30.5)	0.192	2.8 (1–8)	0.049
Both invasions vs no invasion	4.3 (1.4–13.6)	0.014	3.9 (0.6–27.8)	0.169	4 (1.5–10.9)	0.007
Resection edge Not reported vs Clean (>5 mm)	4.5 (1.5–13.6)	0.008	8.3 (0.7–96.1)	0.089	3.1 (1.2–8)	0.019
Resection edge Close (<5 mm) vs Clean (>5 mm)	1.9 (0.7–4.7)	0.190	5 (0.6–42.1)	0.143	1.5 (0.6–3.7)	0.345
LN yield 1–10 vs 0	11.6 (1.2–9.7)	0.033	-	-	10 (1.8–54.2)	0.008
LN yield 11–20 vs 0	5.3 (0.6–46.9)	0.134	-	-	2.7 (0.6–13.4)	0.216
LN yield 21+ vs 0	4.7 (0.6–40.1)	0.156	-	-	5 (1.1–23.7)	0.041
FC_miR-21	1.1 (1–1.2)	0.011	-	-	1.1 (1–1.2)	0.002
FC miR-106b *	-	-	1.3 (0.8–1.9)	0.269	-	-

* for the DSS model, only miR-106b was significant in univariate models, thus this miRNA was selected for multivariate models despite reducing the total number of cases evaluated to 69 due to missing data for some cases. HR—hazard ratio, CI—confidence interval, Sm—smoker, NSmND—never smoker/never drinker, SmD—smoker and drinker, LN—lymph node, FC—fold change, NA—not applicable.

## Data Availability

The data presented in this study are available in the Appendix A.

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
