# Peer review of "Head and Neck Cancer Patients’ Survival According to HPV Status, miRNA Profiling, and Tumour Features—A Cohort Study"

_ijms, 2023, doi:10.3390/ijms24043344_

Round 1
Reviewer 1 Report
The proposed MS entitled Head and neck cancer patients’ survival according to HPV status, miRNA profiling, and tumor features focuses on the attempt to correlate tumors characteristics, HPV status, and miRNAs, and identify their association with patients’ outcomes.
In the first sentence (lines 34-37) authors referenced the article from 2016. I strongly believe that the authors can use the newest one. It was published seven years ago.
The major issue is the authors wrote in the methods part: `type specific primers as described previously (16)`. Reference no. 16 transfers to another paper published by Božinović et al. Where the authors wrote, ´The presence of HPV DNA was assessed using three types of consensus and one type specific PCR primer pair as described previously´. This sentence is referenced to no. 23. Once again, the audience has to open a new manuscript, which was published in 2007. Then, in this paper, the authors wrote: Cervical DNA samples were tested for the presence of HPV DNA by PCR-based method as previously described.
Due to the lack of this information in the original (first) paper and in the second and third, and so on, I recommend rejecting the proposed manuscript without further reading. This part must be improved!
Author Response
REVIEWER 1
The proposed MS entitled Head and neck cancer patients’ survival according to HPV status, miRNA profiling, and tumor features focuses on the attempt to correlate tumors characteristics, HPV status, and miRNAs, and identify their association with patients’ outcomes.
In the first sentence (lines 34-37) authors referenced the article from 2016. I strongly believe that the authors can use the newest one. It was published seven years ago.
Response:
We agree that the data and reference might be a bit too old so we accessed the most recent published GLOBOCAN data (2020 version) and recalculated the incidence (by combining the lip, oral cavity, oropharynx, nasopharynx, hypopharynx, and larynx subsites) as well as updated the reference citations to the most recent Nature review disease primer paper on this topic at lines 34-39 of the revised manuscript
Sung H, Ferlay J, Siegel RL, Laversanne M, Soerjomataram I, Jemal A, Bray F. Global cancer statistics 2020: GLOBOCAN estimates of incidence and mortality worldwide for 36 cancers in 185 countries. CA Cancer J Clin. 2021 Feb 4. doi: 10.3322/caac.21660
Johnson, D.E., Burtness, B., Leemans, C.R. et al. Head and neck squamous cell carcinoma. Nat Rev Dis Primers 6, 92 (2020). doi:10.1038/s41572-020-00224-3
The major issue is the authors wrote in the methods part: `type specific primers as described previously (16)`. Reference no. 16 transfers to another paper published by Božinović et al. Where the authors wrote, ´The presence of HPV DNA was assessed using three types of consensus and one type specific PCR primer pair as described previously´. This sentence is referenced to no. 23. Once again, the audience has to open a new manuscript, which was published in 2007. Then, in this paper, the authors wrote: Cervical DNA samples were tested for the presence of HPV DNA by PCR-based method as previously described.
Response:
We also agree that the train of back citations is now a bit too long to easily access. To fix this issue we now provide a comprehensive Supplementary methods document (referenced at line 129) that provides the details of the PCR methods we continuously adapted for the last two decades
Due to the lack of this information in the original (first) paper and in the second and third, and so on, I recommend rejecting the proposed manuscript without further reading. This part must be improved!
Response:
We fully understand the reviewer’s frustration due to the previous omissions in methods description and we hope that the newly provided detailed methods description as well as the most recent background information will serve to avoid such problems for all readers.
Reviewer 2 Report
The authors assessed the role of miRNA panel in head and neck cancer in association with other clinical and pathological features. The abstract was clear and comprehensive.
Introduction:
Reference for line 35 is warranted.
Lines 61 to 67 highlights the importance of LNY and LNR. While the authors referred to reference #9. Wondering if the level of LN in the neck would be more important rather than the count of the LN positive/biopsied since it is subjective based on the surgeon and the degree of resection.
Line 68: Being a complex disease arising due to accumulation of genomic mutations in addition to epigenomic and transcriptomic alterations, I am not sure if the phrase would sound accurate for all H&N cancers.
Statistical analysis:
Line 159: "non-redundant variables" The term is unclear. Is it the significant variables in univariate regression model were selected for multivariate model?
Results:
A good remark to measure the kappa value for agreements, rarely considered.
Fig. 1 is difficult to be understood being black and white with dots similar to each other. Need to be colored based on miRNA. Was there a significant difference between sample groups?
Overall survival section: suggest using max 3 digits only for p values.
Kaplan-Meier curves: Expecting to plot curves based on miRNA expression at median or Youden index point (high expressor vs low expressor) without integration with staging.
Also suggest performing ROC analysis of single miRNA and miRNA panel.
miRNA risk score could also be generated using beta coefficient of Cox regression and this single value be evaluated for prognosis and survival.
Suggest rephrasing the conclusion to be more consistent and inclusive to the whole story and results.
Author Response
We thank the reviewer for detailed and constructive feedback.
Due to the nature of our responses which included calculations and figures we uploaded the answers as a separate PDF attachment.

Reviewer 3 Report
Dear authors, this is a well designed study, with a meticulous methodology and well presentation of results. Some minor points need improvement, according to STROBE guidelines (eg. type of manuscript in title). Overall, a publishable-worthy study, with a significant clinical importance in the field of head and neck cancer.
Author Response
REVIEWER 3
Dear authors, this is a well designed study, with a meticulous methodology and well presentation of results. Some minor points need improvement, according to STROBE guidelines (eg. type of manuscript in title). Overall, a publishable-worthy study, with a significant clinical importance in the field of head and neck cancer.
Response:
We thank the reviewer for encouraging feedback.
Furthermore, as suggested by the reviewer we have downloaded and consulted the STROBE checklist for cohort studies and adapted the title and text at few places to explicitly include the cohort study design.